# Meaningful Learning: Enhancing Abstract Reasoning in Large Language Models via Generic Fact Guidance

**Kai Xiong**♠ **Xiao Ding**♠* **Ting Liu**♠ **Bing Qin**♠
**Dongliang Xu**♦ **Qing Yang**♦ **Hongtao Liu**♦ **Yixin Cao**♣*

♠Research Center for Social Computing and Information Retrieval
Harbin Institute of Technology, Harbin, China
♣Fudan University, Shanghai, China
♦Du Xiaoman (Beijing) Science Technology Co., Ltd., Beijing, China
{kxiong, xding, tliu, qinb}@ir.hit.edu.cn
{xudongliang, yangqing, liuhongtao01}@duxiaoman.com
yxcao@fudan.edu.cn

## Abstract

Large language models (LLMs) have developed impressive performance and strong explainability across various reasoning scenarios, marking a significant stride towards mimicking human-like intelligence. Despite this, when tasked with several simple questions supported by a generic fact, LLMs often struggle to abstract and apply the generic fact to provide consistent and precise answers, revealing a deficiency in abstract reasoning abilities. This has sparked a vigorous debate about whether LLMs are genuinely reasoning or merely memorizing. In light of this, we design a preliminary study to quantify and delve into the abstract reasoning abilities of existing LLMs. Our findings reveal a substantial discrepancy between their general reasoning and abstract reasoning performances. To relieve this problem, we tailor an abstract reasoning dataset (AbsR) together with a meaningful learning paradigm to teach LLMs how to leverage generic facts for reasoning purposes. The results show that our approach not only boosts the general reasoning performance of LLMs but also makes considerable strides towards their capacity for abstract reasoning, moving beyond simple memorization or imitation to a more nuanced understanding and application of generic facts. The code is available at https://github.com/Waste-Wood/MeanLearn.

## 1 Introduction

Humans proceed with meaningful learning to induce common patterns or high-level abstractions to acquire abstract reasoning abilities [29, 37]. Such capabilities allow us to apply broad principles across diverse situations, demonstrating a deep understanding and versatile application of knowledge. As shown in Figure 1 (a), humans can deduce "rock dissolved" and "the skin suffers pain" when given "adding rock into hydrochloric acid" and "acid touches human skin", respectively. This stems from the established expertise in the human mind (the generic fact "acid is corrosive"), and extends to applications in different scenarios. Of late, remarkable headways of LLMs have pushed AI much further towards human-like intelligence [2, 47]. Interestingly, can LLMs act like humans to instinctively consider the generic fact for flexible applications in different scenarios?

In our pilot investigation, we observed that LLMs appear to lack satisfying capabilities in abstract reasoning. As illustrated in Table 1, there is a notable discrepancy—exceeding 17%—between vanilla

---

*Corresponding Authors

38th Conference on Neural Information Processing Systems (NeurIPS 2024).

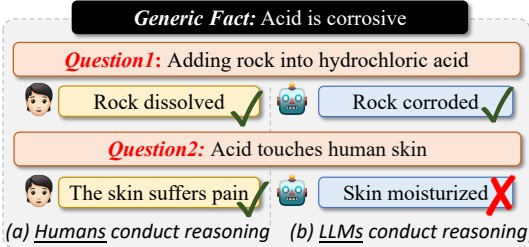

(a) _Humans_ conduct reasoning  (b) _LLMs_ conduct reasoning

Figure 1: The responses of (a) Humans and (b) LLMs when facing two questions which are supported by the same generic fact.

Table 1: The vanilla accuracy and abstract reasoning accuracy (`AbsAcc`) on e-CARE [14]. We will formally define `AbsAcc` in Sec. 2.1.

| Methods | Size | Vanilla Accuracy | AbsAcc |
|---------|------|------------------|--------|
| LLaMA-2 [47] | 7B | 51.67 | 24.82 |
| | 13B | 68.45 | 42.94 |
| Orca-2 [35] | 7B | 74.23 | 51.19 |
| | 13B | 79.83 | 59.47 |
| GPT-3.5 [7] | >20B | 83.25 | 66.08 |
| Human [14] | – | 92.00 | 89.90 |

accuracy and abstract reasoning accuracy (`AbsAcc`) for all LLMs, while humans show only a small disparity. Despite the extensive pre-training of LLMs, which equips them with a vast repository of generic facts, they seem unable to leverage this information as flexibly as humans do (Figure 1 (b)).

In this paper, we aim to systematically and quantitatively investigate the abstract reasoning of LLMs. We first formally define the abstract reasoning metric. Then we carry out abstract reasoning and knowledge probing tasks for a preliminary study. Our investigation comprises two main components: the reasoning tasks can quantify the abstract reasoning capabilities of LLMs. The generic fact probing task can give a further analysis from the perspective of generic fact mastery. Hereafter, we improve abstract reasoning in LLMs based on the analysis of the preliminary study. Unlike existing chain-of-thought (CoT) methods [52, 27], which conduct reasoning with uncontrolled step-by-step explanation, we create a dataset AbsR tailored for abstract reasoning. AbsR not only includes generic facts but also their guided explanations, offering a coached approach to understand the process of reasoning with common patterns. Finally, to make LLMs subconsciously exploit the generic fact like humans, we design a simple but effective learning paradigm called meaningful learning (MeanLearn) to simulate the implicit knowledge learning process [13], enabling LLMs to implicitly learn and utilize generic facts for reasoning without requiring generic facts as input.

We evaluate MeanLearn on six out-of-domain (OOD) reasoning and language understanding benchmarks. Experimental results demonstrate that MeanLearn not only improves general reasoning of LLMs but also excels in abstract reasoning. This distinction in performance is particularly pronounced in the realm of abstract reasoning, underscoring the unique efficacy of our approach in nurturing the higher-order thinking skills that are essential for sophisticated cognitive processing in artificial intelligence systems. Further analysis and ablation studies yield additional evidence supporting the effectiveness of our approach. We summarize our contributions as follows:

- We provide a systematic and quantitative analysis of abstract reasoning in current LLMs, and we develop an abstract reasoning dataset with generic-fact-guided explanations.
- We propose meaningful learning to improve abstract reasoning in LLMs with generic fact.
- We achieve significant improvement in general reasoning and abstract reasoning on various OOD reasoning and language understanding benchmarks.

## 2 Abstract reasoning study

In this section, we initially define the evaluation metric of abstract reasoning to establish quantifiable standards. Then we introduce the LLMs used in this study. Finally, we conduct two experiments: one to directly estimate the abstract reasoning abilities of LLMs, and the other to further analyze it from the perspective of general fact mastery.

### 2.1 Abstract reasoning metric

**Abstract reasoning** requires models to apply general patterns of high-level abstractions to different scenarios or questions. Different from general reasoning, which focuses on each single example, abstract reasoning takes the examples supported by a generic fact as a whole. We define abstract reasoning accuracy due to the lack of a proper metric.

Table 2: Accuracy of generic fact probing.

| LLMs | LLaMA-2 | | Orca-2 | | GPT-3.5 |
|------|---------|-----|--------|-----|---------|
| | 7B | 13B | 7B | 13B | >20B |
| **Accu.** | 62.44 | 70.32 | 26.97 | 12.78 | 77.30 |

Table 3: The categorized abstract reasoning accuracy based on whether the generic facts are known.

| Category | Metrics | LLaMA-2 | | Orca-2 | | GPT-3.5 |
|----------|---------|---------|-------|--------|-------|---------|
| | | 7B | 13B | 7B | 13B | >20B |
| **Known** | AbsAcc | 34.26 | 43.51 | 64.98 | 81.51 | 68.31 |
| **Unknown** | AbsAcc | 17.19 | 31.84 | 49.00 | 58.38 | 58.49 |

Given a dataset $D$ with $n$ generic facts $R = \{r_1, r_2 \cdots, r_n\}$, for the $i$-th generic fact $r_i$, it supports $m_i$ examples $S_i = \{s_1^i, s_2^i, \cdots, s_{m_i}^i\} \in D$.

As shown in Figure 2, akin to Qiu et al. [37], we suppose an LLM $\mathcal{LM}$ has grasped the generic fact $r_i$ if and only if $\mathcal{LM}$ can correctly answer all examples in $S_i$. Hence, we define abstract reasoning accuracy (denoted as AbsAcc) as the proportion of generic facts that $\mathcal{LM}$ has grasped:

$$\text{AbsAcc} = \frac{\sum_i \mathbb{I}(\Phi(\mathcal{LM}, S_i) = |S_i|)}{n}, \quad (1)$$

where $\Phi$ quantifies the number of examples in $S_i$ that $\mathcal{LM}$ answers correctly. $\mathbb{I}$ serves as the indicator function, taking 1 when the specified condition is met and 0 otherwise.

Figure 2: Computation of abstract reasoning metric.

## 2.2 Large language models for evaluation

We employ several LLMs in our preliminary experiments: (1) **LLaMA-2** [47] is an open-access LLM. We use 7B and 13B LLaMA-2 for experiments; (2) **Orca-2** [35] is an open-access LLM finetuned on LLaMA-2 with over 80K reasoning-specific examples. We use 7B and 13B Orca-2 for experiments. (3) **GPT-3.5** [7] is a limited access LLM, we use gpt-3.5-turbo-0125 for experiments.

## 2.3 Experiment I: abstract reasoning

We formulate the first preliminary experiment to gain an initial insight into the abstract reasoning abilities of current LLMs.

To achieve this goal, we choose a multiple-choice question-answering dataset e-CARE [14] for implementation. e-CARE is a large-scale explainable causal reasoning dataset with a generic fact in each example, and a generic fact can support more than one example. The full set (train, test, and dev) of e-CARE is selected for experiments. We filter out the examples that the corresponding generic facts support only one example. Finally, we obtain 13,045 examples supported by 5,608 generic facts.

We follow zero-shot setting to prompt the LLMs for answer selection from multiple choices, and the generic facts are not provided in the inputs (refer to Appendix B.1). Table 1 shows the overall results:

(1) The significant disparity (over 17%) between the vanilla accuracy and AbsAcc for all LLMs proves two aspects: on the one hand, vanilla accuracy is not ideal for estimating abstract reasoning. On the other hand, although LLMs could answer questions accurately, they still cannot capture the common patterns or generic facts behind the questions.

(2) Larger LLMs outperform smaller ones due to their richer knowledge and stronger reasoning ability. Orca-2, additional trained on LLaMA-2, notably outperforms LLaMA-2, indicating that supervised fine-tuning enhances both reasoning and abstract reasoning abilities. However, even with large-scale post-training, the significant gap between vanilla accuracy and AbsAcc remains unbridged.

(3) The minimal gap between vanilla accuracy and AbsAcc for humans suggests the large gap for LLMs is not due to varying difficulties of the example supported by the same generic fact.

| **Generic Fact:** Unusual conditions affect behavior | |
|---|---|
| ***Question:*** Which of the following scenarios is most likely to influence a person's typical shopping habits? | ***Question:*** In a psychological study, which condition is most likely to yield atypical results in participants? |
| ***Options:***
A) A regular day with no significant changes in weather or social conditions
B) A holiday season with sales and promotions in stores
C) When the person has not received their paycheck yet
D) A day with consistent weather and social conditions as the previous week | ***Options:***
A) A study conducted during a stressful event
B) A study conducted in a controlled environment with no distractions
C) A study conducted with the same participants as a previous similar study
D) A study conducted with used questionnaires |
| ***Answer:*** B) | ***Answer:*** A) |
| ***Explanation:*** Holiday seasons with promotions and sales present unusual conditions that can significantly alter a person's typical shopping behavior, often encouraging more spending. | ***Explanation:*** A stressful event like a natural disaster creates an unusual condition that can significantly affect participants' behavior and responses, leading to atypical results in a psychological study. |

Figure 3: Two samples of AbsR guided by the generic fact about "unusual conditions".

## 2.4 Experiment II: generic fact probing

To delve into the abstract reasoning from the perspective of generic fact mastery, we design a generic fact probing task to ask LLMs whether they possess the given generic fact (refer to Appendix B.2 for more details). Drawing inspiration from knowledge uncertainty estimation [3], we formulate this task as a yes or no problem. The overall results are shown in Table 2. We could conclude as follows:

(1) LLaMA-2 and GPT-3.5 have substantial knowledge reserves. This indicates the poor `AbsAcc` of LLaMA-2 might come from a lack of ability to apply the generic fact to different reasoning scenarios. This problem also exists in GPT-3.5, but it is not as serious as LLaMA-2.

(2) In contrast, Orca-2 knows limited generic facts, and the larger the model, the less it knows. This contrasts sharply with its strong reasoning abilities (vanilla accuracy). We suppose training Orca-2 is more like rote learning rather than meaningful learning, which implies that Orca-2 tends to lose knowledge while acquiring reasoning.

To deeply investigate the relation between generic facts and abstract reasoning, we categorize the examples in e-CARE according to whether LLMs know the generic facts. The results are represented in Table 3, from which we can draw the following conclusions:

(1) The performance of "Known" category has superiority over "Unknown" category across all LLMs. This demonstrates knowing the generic facts is helpful for abstract reasoning, which provides a potential avenue for abstract reasoning enhancement through generic fact retrieval.

(2) Nonetheless, most of the results in the "Unknown" category still have undeniable performance, suggesting that LLMs may engage in reasoning based on suspicious correlations or rely on memorization.

In conclusion, we can enhance the abstract reasoning abilities of LLMs in two ways: (1) ***Knowledge***, injecting generic facts through additional training or prompting; (2) ***Reasoning***, teaching LLMs how to utilize the generic facts to provide more precise responses to questions.

## 3 AbsR: abstract reasoning dataset

To enhance the abstract reasoning abilities of LLMs through ***Knowledge*** and ***Reasoning*** ways, we need a dataset that can not only provide generic facts but also teach LLMs how to exploit the generic fact to derive the correct answer in different scenarios (meaningful learning). Since there is no suitable dataset to assist us in achieving the goals, we create an **Abs**tract **R**easoning dataset (AbsR).

### 3.1 Generic fact collection

To obtain the generic facts, We chose GenericsKB [5] as the foundational generic fact base for AbsR. GenericsKB is a large-scale knowledge base with over 3.4 million sentences (e.g. Dogs bark)

expressing general truths, each of which also includes the corresponding concept (e.g. Dog) and the confidence score (between 0 and 1). Hence, we filter and then sample a collection of 4,613 high-quality generic facts of different concepts from GenericsKB (details of generic fact filtering and sampling can refer to Appendix C).

## 3.2 Dataset construction

To construct the whole dataset, and inspired by previous works [49, 56], we choose GPT-4 [2] as our data annotator. The API we used is `gpt-4-1106-preview`.

Table 4: The statistics of AbsR.

|       | Examples | Questions | Generic Facts |
|-------|----------|-----------|---------------|
| Train | 18,020   | 9,010     | 4,613         |
| Test  | 200      | 200       | 104           |

Specifically, for each sampled generic fact $r_i$, we would ask GPT-4 to create samples $S_i = \{s_1^i, \cdots, s_{m_i}^i | 1 \leq m_i \leq 3\}$ in various scenarios based on $r_i$. Each sample $s_j^i$ contains the following terms: (1) a question $X_j^i$ with a few options, (2) a response $Y_j^i$ with an answer and an explanation guided by $r_i$. All terms form a triple $s_j^i = < X_j^i, r_i, Y_j^i >$ for each sample. The prompt for dataset creation can refer to Appendix D. Figure 3 shows an illustration of the created samples.

Finally, without loss of generality, given the $j$-th sample $s_j^i$ of generic fact $r_i$, we can create two kinds of examples for meaningful learning. One is predicting $Y_j^i$ given $< X_j^i, r_i >$ (K-example), while the other is predicting $Y_j^i$ given only $X_j^i$ (R-example). The examples can implicitly enhance abstract reasoning in LLMs through ***Knowledge*** and ***Reasoning*** ways. Table 4 shows the statistics of AbsR.

## 3.3 The quality of AbsR

We conduct human evaluations to measure the quality of AbsR from the following dimensions (details and more statistics such as evaluation criteria, agreements, and pay can refer to Appendix E and F):

- **Human Performance**: humans can achieve vanilla accuracy of 95% and `AbsAcc` of 93.27%;
- **Support Rate**: 89% (the rate of examples which can be supported by the generic fact);
- **Diversity**: 88.5% (the diversity of the examples supported by the same generic fact. Greater diversity indicates greater differences among the examples of the same generic fact).

For comparison, e-CARE [14] is also human-annotated, with examples generated from generic facts. On e-CARE, humans reached 92% vanilla accuracy, 89.9% `AbsAcc`, and an 87% support rate [14], showing that AbsR matches the quality of the human-annotated dataset.

## 4 Method: meaningful learning

To imitate humans' instinctive use of general facts in reasoning, we develop a simple but effective learning paradigm called meaningful learning (MeanLearn). It can enhance the abstract reasoning abilities of LLMs in ***Knowledge*** and ***Reasoning*** ways. This is inspired by the implicit knowledge learning process, which uses hidden variables to learn event background knowledge [13]. MeanLearn can make LLMs implicitly learn generic facts and solve problems under the guidance of generic facts.

Specifically, an LLM $\mathcal{LM}$ with parameters $\theta$ and the input $x$ can construct a conditional probability for the output $y$:

$$\mathcal{LM}(x, y, \theta) = -\sum_t \log p_\theta(y_t | x, y_{<t}). \tag{2}$$

In MeanLearn, for a K-example $< X, r, Y >$ and R-example $< X, Y >$ pair guided by generic fact $r$, we send the example pair into $\mathcal{LM}$ to model two conditional probabilities:

$$\mathcal{LM}(X, Y, \theta) = -\sum_t \log p_\theta(Y_t | X, Y_{<t}),$$
$$\mathcal{LM}(X, r, Y, \theta) = -\sum_t \log q_\theta(Y_t | X, r, Y_{<t}). \tag{3}$$

Hereafter, on the one hand, we train $\mathcal{LM}$ to learn policies $p_\theta(Y_t | X, Y_{<t})$ and $q_\theta(Y_t | X, r, Y_{<t})$ together to enable $\mathcal{LM}$ implicitly learn the generac fact $r$. On the other hand, $\mathcal{LM}$ can learn how

to apply $r$ into different scenarios by learning the explanations in $Y$ of different $X$ supported by $r$. Finally, we can reason with the implicit guidance of $r$ when only given $X$, just like humans instinctively answer questions without explicitly giving the general facts.

## 5 Experiments

### 5.1 Experimental details

**Training setup.** We use LoRA [20] for parameter effcient finetuning. 7B and 13B MeanLearn are trained on 7B and 13B Orca-2, respectively, 8B MeanLearn is trained on 8B LLaMA-3, with batch sizes of 256 for 7B and 8B, and 240 for 13B. MeanLearn of various sizes are trained for one epoch at a 5e-5 learning rate with AdamW [26]. 7B and 8B MeanLearn are trained on 2 NVIDIA A100 80GB PCIe GPUs for 3 hours. 13B MearnLearn is trained on 3 such GPUs for 4 hours.

**Baselines.** We adopt a wide range of open and limited access LLMs across different sizes as baselines. **Open Access LLMs**: (1) LLaMA-2 [47] 7B and 13B; (2) LLaMA-3 [34] 8B; (3) Vicuna [8] 7B and 13B (finetuned on ShareGPT [1] data); (4) WizardLM [56] 7B and 13B (fintuned with evolved instruction data); (5) Orca-2 [35] 7B and 13B (progressively finetuned on massive reasoning data). **Limited Access LLMs**: (1) GPT-3.5 (`gpt-3.5-turbo-0125`); (2) PaLM-2 (`text-bison-001`).

**Evaluation benchmarks.** In addition to AbsR, we also include a wide range of benchmarks of reasoning and natural language understanding for evaluation: (1) AGIEval [60] consists of tests ranging from college admission tests, to national civil service examinations; (2) RACE [28] consists of tests with reading comprehension questions; (3) BBH [43] is a subset of Big-Bench [42], which contains 23 hardest tasks focusing on challenging scenarios; (4) Com. [55] is a collection of 7 commonsense reasoning datasets ($\alpha$NLI [4], CSQA [44], COPA [39], e-CARE [14], SocialIQa [40], PIQA [6], and StrategyQA [16]); (5) MMLU [18] is a massive multitask language understanding benchmark; (6) ARC [10] is a benchmark of easy (ARC-e) and challenge (ARC-c) science questions.

**Evaluation setup.** Since generation-based evaluation is time-consuming due to limited computing resources, we follow OpenCompass [11] to have hybrid evaluation criteria with high reproducibility: (1) For PaLM-2 and GPT-3.5 on all tasks, and open access LLMs on 4 generation tasks (4 tasks in BBH), greedy decoding and exact match are utilized for evaluation; (2) For open access LLMs on classification and multiple-choice tasks, perplexity (PPL) is adopted for prediction. The option or category with the lowest PPL is chosen as the answer. For all baselines, evaluation is conducted once since there is no randomness on such evaluation criteria. For MeanLearn, results are averaged of MeanLearn(s) trained with 3 different random seeds. The evaluation prompts can refer to Appendix H.

### 5.2 Results: vanilla accuracy

Table 5 shows the `Vanilla Accuracy` of MeanLearn and baselines:

(1) Additional training usually brings improvements (Vicuna, WizardLM, Orca-2, and MeanLearn), but it still can result in performance losses on some benchmarks (e.g. AGIEval). This signifies the effectiveness of massive reasoning-specific data and the necessity of data coverage.

(2) MeanLearn has superiority over all open access baselines on nearly all benchmarks. This demonstrates that MeanLearn can use generic facts to effectively guide LLMs to reason more properly and logically, making them more flexible and task-adaptive.

(3) The more challenging the benchmark (e.g. BBH), the less pronounced the improvement in the performance of MeanLearn. This is mainly because the improvement is mostly constrained by the complex scenarios in harder benchmarks, which require mightier base LLMs or more complex training data to perform and learn multi-step and compositional reasoning.

(4) MeanLearn does not have many advantages over LLaMA-3. LLaMA-3 stores dense knowledge in its parameters, making it harder to learn new knowledge without forgetting existing knowledge.

(5) Larger open-sourced LLMs do not always yield better results. We suppose the main factors behind this are in two ways: on the one hand, LLMs with different parameter sizes might be good at different tasks, and excessive contemplation may lead to decreases on some tasks [53]. On the other hand, the perplexity-based evaluation setting might have moderate cross-scale generalization ability.

Table 5: Overall vanilla accuracy and `AbsAcc` (%) of baselines and MeanLearn. Due to the space limit, we only report the standard deviation of *Average* performance of each method.

| Size | LLMs | AbsR | AGIEval | RACE | BBH | Com. | MMLU | ARC-e | ARC-c | *Average* |
|------|------|------|---------|------|-----|------|------|-------|-------|-----------|
| `Vanilla Accuracy` | | | | | | | | | | |
| >20B | PaLM-2 | 75.76 | 15.83 | 75.82 | 48.73 | 78.05 | 58.73 | 89.95 | 82.37 | 65.77 ± 0.00 |
| | GPT-3.5 | 84.00 | 25.84 | 83.36 | 54.15 | 74.80 | 65.98 | 94.71 | 88.81 | 71.46 ± 0.00 |
| 7B | LLaMA-2 | 50.00 | 27.89 | 38.85 | 26.00 | 55.37 | 41.65 | 58.73 | 43.05 | 42.69 ± 0.00 |
| | Vicuna | 75.00 | 32.56 | 65.75 | 31.93 | 64.77 | 49.40 | 74.78 | 57.63 | 56.48 ± 0.00 |
| | WizardLM | 65.00 | 22.27 | 22.11 | 25.99 | 43.63 | 23.26 | 25.57 | 22.37 | 31.28 ± 0.00 |
| | Orca-2 | 73.50 | **38.57** | 73.32 | 34.51 | 71.58 | 50.11 | 79.19 | 73.90 | 61.84 ± 0.00 |
| | MeanLearn | **77.00** | 38.15 | **77.15** | **35.64** | **76.59** | **52.98** | **86.67** | **78.06** | **65.28** ± 0.34 |
| 8B | LLaMA-3 | 82.50 | 36.33 | 76.75 | 36.60 | **71.76** | 61.86 | 88.54 | 75.25 | 66.20 ± 0.00 |
| | MeanLearn | **84.50** | **38.53** | **77.02** | **37.67** | 71.73 | **62.35** | **88.89** | **77.97** | **67.12** ± 0.18 |
| 13B | LLaMA-2 | **73.50** | 34.41 | 59.46 | 30.61 | 61.00 | **51.87** | 71.25 | 55.25 | 54.67 ± 0.00 |
| | Vicuna | 74.00 | 33.23 | 63.54 | 33.22 | 63.10 | 49.34 | 77.78 | 59.77 | 56.75 ± 0.00 |
| | WizardLM | 74.00 | 33.23 | 63.54 | 33.22 | 63.10 | 49.34 | 77.78 | 59.77 | 56.75 ± 0.00 |
| | Orca-2 | 66.50 | 43.75 | 66.86 | 39.39 | 65.92 | 47.27 | 82.19 | 70.85 | 60.34 ± 0.00 |
| | MeanLearn | 67.17 | **43.92** | **70.24** | **40.36** | **69.75** | 51.68 | **88.69** | **82.00** | **64.23** ± 0.25 |
| `AbsAcc` | | | | | | | | | | |
| >20B | PaLM-2 | 64.58 | 9.59 | 61.54 | 27.69 | 69.34 | 44.39 | 85.68 | 77.10 | 54.99 ± 0.00 |
| | GPT-3.5 | 77.08 | 14.48 | 66.79 | 25.45 | 65.74 | 52.09 | 92.16 | 85.05 | 59.86 ± 0.00 |
| 7B | LLaMA-2 | 35.42 | 16.09 | 14.08 | 9.56 | 47.42 | 25.48 | 50.27 | 35.51 | 29.23 ± 0.00 |
| | Vicuna | 58.33 | 19.90 | 38.68 | 14.28 | 58.07 | 32.50 | 65.95 | 49.53 | 39.84 ± 0.00 |
| | WizardLM | 53.24 | 12.94 | 5.57 | 9.77 | 32.96 | 11.55 | 20.00 | 17.29 | 20.42 ± 0.00 |
| | Orca-2 | 60.42 | **26.27** | 50.90 | 15.67 | 64.02 | 33.68 | 72.16 | 67.76 | 48.86 ± 0.00 |
| | MeanLearn | **64.58** | 25.27 | **57.10** | **16.67** | **71.17** | **37.24** | **81.35** | **73.83** | **53.39** ± 0.32 |
| 8B | LLaMA-3 | 72.92 | 23.78 | **55.61** | 18.13 | 66.35 | **47.31** | 83.51 | 70.09 | 54.71 ± 0.00 |
| | MeanLearn | **73.96** | **26.57** | 55.09 | **19.24** | **66.48** | 46.76 | **83.78** | **73.36** | **55.66** ± 0.17 |
| 13B | LLaMA-2 | **61.46** | 21.01 | 32.17 | 13.74 | 52.87 | **35.72** | 62.43 | 44.86 | 40.53 ± 0.00 |
| | Vicuna | 61.46 | 19.70 | 36.50 | 16.33 | 55.26 | 32.21 | 70.27 | 52.34 | 40.37 ± 0.00 |
| | WizardLM | 59.38 | 16.96 | 28.59 | 18.61 | 55.73 | 29.47 | 57.84 | 42.06 | 38.58 ± 0.00 |
| | Orca-2 | 50.00 | 29.50 | 42.12 | 20.61 | 56.41 | 31.11 | 75.95 | 62.15 | 45.98 ± 0.00 |
| | MeanLearn | 45.82 | **30.01** | **47.65** | **21.85** | **61.04** | 35.21 | **85.41** | **77.08** | **50.51** ± 0.23 |

(6) LLMs possess more than 20B parameters still struggle with abstract reasoning. MeanLearn can achieve comparable performance to PaLM-2 despite their huge gap in model sizes.

## 5.3 Results: abstract reasoning accuracy

Since the generic facts are not given in the OOD benchmarks, we train a RoBERTa-Large [32] with constrastive learning to cluster examples supported by the same generic fact. In particular, we utilize e-CARE as the training set, examples guided by the same generic fact will be pushed closer, or they will be pushed away. Cosine similarity is used to measure the distance between examples (training details can refer to Appendix I). For clustering, examples sharing a similarity above 0.6 are considered to be supported by the same generic fact. Each cluster contains no more than 3 examples.

Hereafter, we calculate the abstract reasoning metrics based on the results of clustering. The overall results are shown in Table 5 (`AbsAcc`), we can have the following observations:

(1) `AbsAcc` is much lower than vanilla accuracy, indicating the large gap between general reasoning and abstract reasoning. More efforts should be made to fill this gap in the future.

(2) MeanLearn can surpass all baselines, usually exhibiting greater advantages in `AbsAcc` than in vanilla accuracy. This suggests that the improvement in vanilla accuracy is more likely due to the enhancement of abstract reasoning abilities. MeanLearn can tell LLMs the generic facts and teach them how to implicitly utilize them in different scenarios, resulting in better abstract reasoning.

Table 6: The overall reasults of ablation studies. "w/o" denotes without.

| Size | Method | AbsR | AGIEval | RACE | BBH | Com. | MMLU | ARC-e | ARC-c | *Average* |
|------|--------|------|---------|------|-----|------|------|-------|-------|-----------|
| `Vanilla Accuracy` | | | | | | | | | | |
| 7B | MeanLearn | **77.00** | 38.15 | **77.15** | **35.64** | **76.59** | **52.98** | **86.67** | **78.06** | **65.28** |
| | w/o Knowledge | 73.50 | 38.89 | 74.10 | 33.80 | 73.28 | 50.97 | 79.72 | 75.25 | 62.44 |
| | w/o Reasoning | 72.50 | 38.37 | 72.60 | 33.39 | 71.68 | 49.55 | 77.25 | 73.22 | 61.07 |
| | w/ AbsR* | 73.50 | **38.53** | 73.14 | 33.65 | 73.57 | 50.96 | 80.60 | 75.25 | 62.40 |
| 13B | MeanLearn | **67.17** | 43.92 | **70.24** | **40.36** | 69.75 | **51.68** | **88.69** | **82.00** | **64.23** |
| | w/o Knowledge | 63.50 | **44.07** | 67.19 | 36.80 | **70.45** | 47.88 | 84.13 | 74.58 | 61.08 |
| | w/o Reasoning | 56.00 | 43.95 | 58.05 | 35.91 | 68.09 | 45.31 | 82.54 | 69.83 | 57.46 |
| | w/ AbsR* | 66.50 | 43.75 | 66.86 | 36.92 | 65.92 | 47.27 | 82.19 | 70.85 | 60.03 |
| `AbsAcc` | | | | | | | | | | |
| 7B | MeanLearn | **64.58** | 25.27 | **57.10** | **16.67** | **71.17** | **37.24** | **81.35** | **73.83** | **53.39** |
| | w/o Knowledge | 61.46 | **26.67** | 52.91 | 14.74 | 66.26 | 34.84 | 72.70 | 69.16 | 49.84 |
| | w/o Reasoning | 59.38 | 26.18 | 49.66 | 14.15 | 64.13 | 33.27 | 69.19 | 67.29 | 47.91 |
| | w/ AbsR* | 61.46 | 26.44 | 52.24 | 14.76 | 55.56 | 34.88 | 74.05 | 69.16 | 48.57 |
| 13B | MeanLearn | 45.82 | 30.01 | **47.65** | **21.85** | 61.04 | 35.21 | **85.41** | **77.08** | **50.51** |
| | w/o Knowledge | 46.88 | 30.43 | 42.87 | 18.83 | **62.95** | 32.10 | 78.11 | 66.82 | 47.37 |
| | w/o Reasoning | 36.46 | **30.81** | 31.19 | 18.26 | 59.70 | 29.12 | 75.68 | 61.68 | 42.86 |
| | w/ AbsR* | **50.00** | 29.58 | 42.12 | 18.82 | 56.41 | 31.11 | 75.95 | 62.15 | 45.77 |

# 6 Further analysis

To further investigate the pros, cons, and effectiveness of MeanLearn, we conduct several analyses and ablation studies. LLaMA-2, Orca-2, and MeanLearn are adopted for analysis.

## 6.1 Performance on different domains

We categorize tasks of MMLU based on their respective domains to demonstrate the pros and cons of MeanLearn for future improvements. The results are shown in Figure 4 with a total of 16 categories:

(1) All LLMs possess poor vanilla accuracy and `AbsAcc` in math and chemistry. We conjecture the primary factors are the complexity of math problems and the shortage of chemistry knowledge.

(2) Compared to Orca-2, MeanLearn possesses considerable advantages in engineering, geography, and physics & astronomy domains. We suppose this is related to the domain coverage of AbsR.

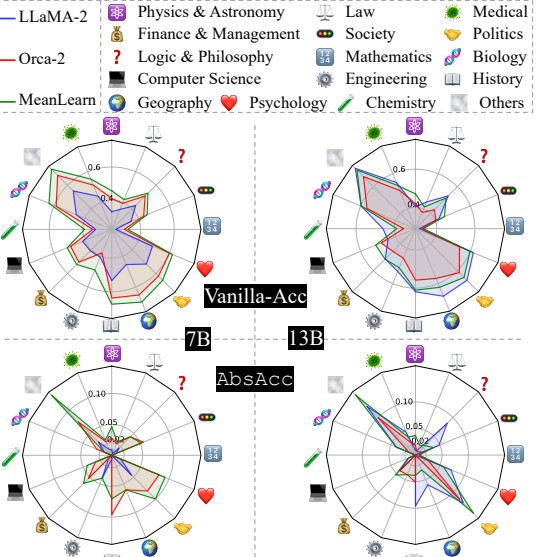

Figure 4: Visualization of reasoning capabilities on MMLU, which is categorized by task domians.

(3) Since the overall `AbsAcc` is not relatively high, for potential improvement in the future, apart from focusing more on the domains with poor vanilla accuracy and `AbsAcc`, we should also increase the overall volume of finetuning data.

## 6.2 How much do knowledge and reasoning ways contribute to MeanLearn?

We utilize *Knowledge* and *Reasoning* ways to enhance abstract reasoning in LLMs. To investigate the effect of them, we conduct an ablation study of AbsR to respectively remove the *Knowledge* (generic facts) and *Reasoning* (explanations) in each example. Results are shown in Table 6 (w/o Knowledge and w/o Reasoning), we can observe:

Table 7: MeanLearn trained based on Mistral.

| Method | AbsR | Com. | MMLU | RACE | Average |
|---|---|---|---|---|---|
| Vanilla Accuracy | | | | | |
| Mistral | 83.00 | 66.53 | 57.02 | 75.33 | 70.47 |
| MeanLearn (Ours) | **84.50** | **74.56** | **58.28** | **76.65** | **73.50** |
| AbsAcc | | | | | |
| Mistral | 72.92 | 58.87 | 40.19 | 52.69 | 56.17 |
| MeanLearn (Ours) | **73.96** | **68.59** | **41.14** | **55.77** | **59.87** |

Table 8: Performance on mathematical reasonin datasets from MMLU [18].

| Size | Method | AA | CM | EM | HSS | HSM | *Average* |
|---|---|---|---|---|---|---|---|
| | Vanilla Accuracy | | | | | | |
| 7B | Orca-2 | 23.00 | 32.00 | 33.07 | 37.95 | 28.15 | 30.83 |
| | MeanLearn | 36.00 | 35.00 | 33.07 | 40.28 | 28.89 | **34.56** |
| | Mistral | 25.00 | 30.00 | 38.89 | 46.76 | 32.96 | 34.72 |
| | MeanLearn | 31.00 | 34.00 | 37.83 | 47.69 | 34.07 | **36.92** |
| | AbsAcc | | | | | | |
| 7B | Orca-2 | 0.00 | 4.88 | 25.00 | 25.17 | 14.63 | 13.94 |
| | MeanLearn | 2.86 | 7.32 | 24.02 | 27.97 | 16.26 | **15.69** |
| | Mistral | 2.86 | 7.32 | 25.49 | 36.36 | 14.63 | 17.33 |
| | MeanLearn | 5.71 | 9.76 | 24.55 | 37.76 | 16.26 | **18.81** |

(1) Removing ***Knowledge*** (generic facts) or ***Reasoning*** (explanation) leads to a decrease in performance. This is because the generic facts and their guided reasoning process in AbsR can teach LLMs to learn and utilize the generic fact behind the questions to reason more logically and properly.

(2) In MeanLearn, ***Knowledge*** contributes less in boosting the vanilla and abstract reasoning than ***Reasoning***. It indicates the importance of teaching LLMs how to reason under the guidance of generic facts. Furthermore, it reveals the bottleneck of small-scale LLMs reasoning is caused more by reasoning itself rather than knowledge.

(3) MeanLearn without Knowledge can defeat Orca-2 on vanilla accuracy and `AbsAcc`, while MeanLearn without Reasoning cannot. This implies the synergy effects between ***Knowledge*** and ***Reasoning*** (MeanLearn) can push LLMs further.

(4) In AGIEval, ***Knowledge*** and ***Reasoning*** are less critical due to their focus on complex reading comprehension without needing vast knowledge. Moreover, ***Knowledge*** seems useless in Commonsense. When providing LLMs with knowledge, we should also teach them how to use it.

## 6.3 Is the improvement in performance solely attributed to additional data?

Supervised fine-tuning can also improve `AbsAcc` (Table 1), so we design another ablation study to investigate whether the improvement of MeanLearn only comes from additional data.

To be precise, for each sample in AbsR, we use GPT-4 to generate a new explanation based on the question and label without the guidance of the generic fact. Hereafter, we obtain AbsR$^*$. The only difference between AbsR and AbsR$^*$ is the explanations in AbsR are guided by the generic fact. Table 6 (w/ AbsR$^*$) shows the results of MeanLearn trained with AbsR$^*$, we can infer:

(1) Training 7B and 13B Orca-2 with AbsR$^*$ can still outperform Orca-2 on vanilla accuracy. However, it cannot bring stable improvements on `AbsAcc`. This underscores that reasoning-specific post-training can boost performance, while improving abstract reasoning performance might depend on more careful design such as MeanLearn.

(2) MeanLearn without Knowledge outperforms MeanLearn trained with AbsR$^*$, showing the superiority of training with generic-fact-guided explanation. This stems from the generic-fact-guided explanation that can teach LLMs to use the regular pattern behind the questions for better reasoning.

## 6.4 Expanding MeanLearn to LLMs besides the LLaMA series

To investigate the universality of MeanLearn, we train MeanLearn with Mistral-7B-Instruct-v0.2 [21] (denoted as Mistral). All settings are consistent with Sec 5. We choose AbsR, Com., MMLU, and RACE for evaluation. As shown in Table 7, we can find MeanLearn has good applicability on Mistral.

## 6.5 Expanding reasoning domains to mathematical reasoning

Although we do not incorporate math examples in AbsR, we want to investigate whether MeanLearn can improve the performance of mathematical reasoning by enhancing abstract reasoning in LLMs.

Following [33], we select five mathematical reasoning datasets from MMLU [18] to demonstrate this: abstract algebra (AA), college mathematics (CM), elementary mathematics (EM), high school statistics (HSS), and high school mathematics (HSM). We choose 7B version of Orca-2 and Mistral for comparison. Results are shown in Table 8, it is interesting to find that MeanLearn can outperform baselines even if there is no math data in AbsR. This demonstrates the effectiveness of MeanLearn in enhancing abstract reasoning in LLMs.

# 7 Related work

## 7.1 Reasoning with LLMs

LLMs have overturned the research paradigm of NLP, which makes reasoning more accessible. Using LLMs to perform reasoning can be categorized into tunning-based and prompt-based methods.

Tunning-based methods, originating from T0 [48] and FLAN [52], finetuned T5 [38] and LaMDA-PT [46] on massive NLP tasks, achieving notable zero-shot performance on unseen tasks. Flan-PaLM [9] and T$k$-INSTRUCT [50] further scaled up the model and task size for improvement. Orca [36] and Orca-2 [35] leveraged vast instruction data to teach small-scale LLMs reasoning skills.

As for prompt-based methods, Wei et al. [53] introduced CoT prompt to elicit reasoning in LLMs by encouraging step-by-step thinking. Additionally, Kojima et al. [27] presented zero-shot CoT to bypass manual annotation issues. Subsequently, Zhang et al. [59] proposed AutoCoT to automatically generate few-shot examples with CoT for reasoning. Various CoT-based methods [22, 51] have since emerged, such as complex CoT [15], and the tree of thought [57], etc.

These methods are designed for improving the general reasoning abilities of LLMs, while we focus on the abstract reasoning capabilities of LLMs.

## 7.2 LLMs-as-annotators

LLMs are increasingly used as annotators to create data for training and evaluation due to their advanced instruction-following skills.

Some studies distilled various CoT data from LLMs to train student models ([30, 19, 58], *inter alia*), achieving impressive performance. For instance, Li et al. [30] distilled symbolic CoT from GPT-3 and observed enhanced performance in commonsense reasoning. Dai et al. [12] used ChatGPT to generate augmentation data to enhance BERT [24]. With the advent of open-source LLMs like LLaMA [47], other studies distilled instruction data from large-scale LLMs to train small-scale models ([56, 36, 25, 31, 23, 17], *inter alia*). For example, Xu et al. [56] and Luo et al. [33] employed an evolving strategy to obtain complex data, trained LLaMA, and achieved strong performance. Tang et al. [45] constructed a concept graph and used GPT-3.5 to synthesize math data with high diversity. We employ LLMs to generate training data and develop AbsR tailored for abstract reasoning.

# 8 Conclusion

In this paper, we investigate the abstract reasoning of existing LLMs. To achieve this, we first define the evaluation metric of abstract reasoning and design a series of preliminary experiments. Then we discover a significant performance gap between general reasoning and abstract reasoning. Hence, we tailor an abstract reasoning dataset AbsR with the help of GPT-4 to enhance LLMs. Finally, we devise a simple but effective paradigm (MeanLearn) to teach LLMs abstract reasoning in ***Knowledge*** and ***Reasoning*** ways. Extensive experiments demonstrate the superiority and effectiveness of MeanLearn in vanilla and abstract reasoning accuracies. The limitations of MeanLearn are discussed in Appendix A.

# Acknowledgements

We would like to acknowledge the editors and reviewers for their efforts and advice, and gratefully acknowledge the support of the National Natural Science Foundation of China under Grants U22B2059 and 62176079, Natural Science Foundation of Heilongjiang Province under Grant YQ2022F005.

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

# A  Limitations

There are several limitations of our work: First, the size of the training data should be larger to obtain better and more stable results. Second, there should be some examples (whether MeanLearn can generate meaningful explanations) to demonstrate the superiority of MeanLearn. Third, more efforts should be made to focus more on abstract reasoning itself. Finally, complex scenarios (with complex generic facts) such as causal chain reasoning [54] and logical reasoning [41] is a potential direction of abstract reasoning.

# B  Asbstract reasoning study

## B.1  Prompt for experiment I

### B.1.1  Orca-2

```
<|im_start|>system
You are Orca, an AI language model created by Microsoft.  You are
a cautious assistant.  You carefully follow instructions.  You are
helpful and harmless and you follow ethical guidelines and promote
positive behavior.<|im_end|>
<|im_start|>user
{premise} Hypothesis1 or Hypothesis2?
Hypothesis1:  {hypothesis1}
Hypothesis2:  {hypothesis2}
Your answer should follow the format like ''Answer:  Hypothesis(1 or 2)
is more plausible.
Explanation:  ___''<|im_end|>
<|im_start|>assistant
```

### B.1.2  LLaMA-2 and GPT-3.5

```
System:  You are a helpful assistant.
User:  {premise} Hypothesis1 or Hypothesis2?
Hypothesis1:  {hypothesis1}
Hypothesis2:  {hypothesis2}
Your answer should follow the format like ''Answer:  Hypothesis(1 or 2)
is more plausible.
Explanation:  ___''
```

The placeholders {premise}, {hypothesis1}, and {hypothesis2} will be filled with the corresponding terms in each example of e-CARE dataset.

### B.2 Prompt for experiment II: Orca-2

#### B.2.1 Orca-2

```
<|im_start|>system
You are Orca, an AI language model created by Microsoft.  You are
a cautious assistant.  You carefully follow instructions.  You are
helpful and harmless and you follow ethical guidelines and promote
positive behavior.<|im_end|>
<|im_start|>user
You are given a fact, do you know this fact?  Just answer Yes or No,
do not give any additional information.
Fact:  {fact} <|im_end|>
<|im_start|>assistant
```

#### B.2.2 LLaMA-2 and GPT-3.5

```
System:  You are a helpful assistant.
User:  You are given a fact, do you know this fact?  Just answer Yes
or No, do not give any additional information.
Fact:  {fact}
```

The placeholders {premise}, {hypothesis1}, {hypothesis1}, and {fact} will be filled with the corresponding terms in each example of e-CARE dataset.

## C Details for generic facts filtering

We apply the following steps to filter generics facts from GenericsKB:

(1) Exclude sentences with confidence $< 0.7$.

(2) Categorize the sentences by the concepts.

(3) Randomly sample 4,613 concepts, and then randomly sample one sentence as the generic fact from the category of each sampled concept.

Hereafter, we can obtain a collection of 4,613 generic facts.

## D Prompt for AbsR construction

We use the following prompt to generate our AbsR examples.

You are an expert in creating questions, you should first offer a question together with some options based on the fact the user gives. Second, you should give an answer and an explanation guided by the given fact. You can propose questions in any area, including but not limited to history, law, medicine, math, science, computer science, psychology, AI, politics, economics, etc. Your response should follow the following format: "Question: _____ Options:_____ Answer: _____ Explanation: _____". NOTE that the fact should be an implicit explanation for obtaining the true answer, which means the fact SHOULD NOT appear explicitly in the questions or the options. The explanations should be short. Please create {number} examples.

The placeholder {number} will be randomly filled with "one", "two" or "three".

# E   Details of human evaluation

We choose three annotators with good backgrounds in textual inference and event reasoning. The whole test set of AbsR is involved in the human evaluation (200 instances). For each annotator, we pay $10 per hour, while in our country, the minimum wage is less than $5 per hour. All annotators agreed to let us use their annotations. The evaluation agreements are 95%, 91%, and 90% for the question answering, generic fact supportance, and diversity tasks, respectively.

# F   Instructions of human evaluation

## F.1   Performance

You are given some questions each with some options, all the questions are about commonsense and are generated by GPT-4. Please choose the most plausible option for each question. Just type your choice (such as (A)) in the Answer column.

## F.2   Support rate

You are given some questions each with some options and a generic fact, all the questions are about commonsense and are generated by GPT-4. Please judge whether the generic can support to answer the question. Just type your choice (Yes or No) in the Answer column.

## F.3   Diversity

You are given some generic facts, each generic fact possesses several samples (each sample consists of a question and a few options). The samples can be supported by the corresponding generic fact. Please judge whether the samples within a generic are similar and are just different in nouns or expressions. If sample 1 and sample 2 are similar and are just different in nouns or expressions, just type a tuple (1, 2) in the Answer column. If there are multiple similar sample pairs, just separate them with a "\n". For example, (1, 2)\n(3, 4).

# G   K-example and R-example

## G.1   K-example

```
<|im_start|>system
You are a cautious assistant.  You carefully follow instructions.
You are given a question, a few options, and a rule, you should
follow the given rule to answer the question.  Your response should
follow the format like ''Explanation:  ___Answer:  ____''<|im_end|>
<|im_start|>user
Fact:  Cookie files are simple text files that can be viewed in
Windows Notepad.
Question:  What type of application can be used to open and view
cookie files on a Windows computer?
Options:
A) Microsoft Excel B) Adobe Photoshop C) Windows Notepad D) 3D
Modeling Software<|im_end|>
<|im_start|>assistant
Explanation:  Windows Notepad is a text editor that is capable of
opening and displaying the contents of simple text files, which is
the format of cookie files.
Answer:  C) Windows Notepad<|im_end|>
```

### G.2 R-example

```
<|im_start|>system
You are a cautious assistant.  You carefully follow instructions.
You are given a question together with a few options, you should
give an explanation first and then answer the question.  Your
response should follow the format like ''Explanation:  ___Answer:
____''<|im_end|>
<|im_start|>user
Question:  What type of application can be used to open and view
cookie files on a Windows computer?
Options:
A) Microsoft Excel B) Adobe Photoshop C) Windows Notepad D) 3D
Modeling Software<|im_end|>
<|im_start|>assistant
Explanation:  Windows Notepad is a text editor that is capable of
opening and displaying the contents of simple text files, which is
the format of cookie files.
Answer:  C) Windows Notepad<|im_end|>
```

## H   Templates for evaluation

### H.1   LLaMA-2, LLaMA-3, and MeanLearn (8B)

```
The following are multiple-choice questions (with answers) about
abstract algebra.

Find the degree for the given field extension Q(sqrt(2), sqrt(3),
sqrt(18)) over Q.
A. 0 B. 4 C. 2 D. 6
Answer:  A
```

### H.2   Orca-2 and MeanLearn (7B and 13B)

```
<|im_start|>system
The following are multiple-choice questions (with answers) about
abstract algebra.<|im_end|>
<|im_start|>user
Question:  Find the degree for the given field extension Q(sqrt(2),
sqrt(3), sqrt(18)) over Q.
Options:
A. 0 B. 4 C.  2 D. 6<|im_end|>
<|im_start|>assistant
Answer:  A<|im_end|>
```

## I   Training details of RoBERTa-Large-based clusterer

We use the RoBERTa-Large [32] released by Meta. The batch size is 256. We use AdamW [26] to optimize the model with a learning rate of 1e-5 for 5 epochs. The computing device is one NVIDIA A100-SXM-64GB. The running time is about 12 minutes.

