# OpenReview forum: "Meaningful Learning: Enhancing Abstract Reasoning in Large Language Models via Generic Fact Guidance"
_NeurIPS.cc/2024/Conference — NeurIPS 2024 poster_

### Official Review · Reviewer_Fwhs · 2024-07-08

**Soundness:** 3
**Presentation:** 4
**Contribution:** 3
**Rating:** 6
**Confidence:** 4

**Summary:**

The study introduced a specialized dataset and learning approach that aimed to enhance LLMs' use of generic facts in reasoning. The results indicate that this methodology not only improved their general reasoning skills but also significantly developed their abstract reasoning abilities, suggesting a shift from mere memorization to a deeper, more nuanced understanding and application of information. The paper is well composed and demonstrates good, consistent performance across several benchmarks.

**Strengths:**

- the concept of AbsAcc is interesting and natural, as demonstrated by the experiments in Table 1, where the small difference between vanilla accuracy and AbsAcc for human subjects highlights that vanilla accuracy alone may not be an adequate metric for assessing abstract reasoning.
- the paper is well-composed and straightforward to comprehend.

**Weaknesses:**

- the experiments currently focus on earlier versions of LLMs, including LLaMA-2 and GPT-3.5-turbo-0125. It is advisable to also include more recent models like LLaMA-3 and GPT-4/4o to ensure the consistency of the performance evaluations.
- some of the terminology used appears complex; for instance, in the memory learning section, the distinction between Knowledge and Reasoning examples (K-example and R-example) seems minimal. In the K-example, when a user poses a question, a supporting fact is cited while the R-example not. Additionally, I'm confused by the definition of abstract reasoning. It looks akin to tasks in commonsenseQA (https://aclanthology.org/N19-1421.pdf). The authors might need to elucidate the differences between their approach and that body of research.

**Questions:**

- see the questions in weakness section

**Limitations:**

- some of the terms mentioned in the paper maybe a little bit fancy and overestimated, please refer to the weakness section.

---

> ### Author Rebuttal · Authors · 2024-08-02
>
> Thanks for your insightful comments, the following is the detailed response:
> # 1. More experiments (Weakness 1)
> We have conducted experiments on LLaMA-3 which is shown in Table 5, our MeanLearn can improve the performance of LLaMA-3 on both vanilla accuracy and AbsAcc. We do not test GPT-4/4o is because: (1) our main focus is to improve small LLMs, which is also crucial and have a broader range of application. They are preferred for deployment on devices like cell phones due to their lower resource demands. (2) GPT-4o is released about a week before the deadline of NeurIPS 2024, leaving insufficient time to conduct experiments with GPT-4o. Furthermore, we are glad to offer more exploration in the future.
>
> # 2. Some of the terminology used appears complex (Weakness 2)
> Your understandings about K-example and R-example are correct. While for abstract reasoning, it is not a specific reasoning type. We can compute the performance of abstract reasoning for any task or benchmark, not just CommonsenseQA. For example, in math problems, abstract reasoning can help estimate the ability of LLMs to use the generic fact “x + y = y + x” to solve a series of problems in various scenarios that rely on this fact. While in commonsense problems, it can help estimate the ability of LLMs to use the generic fact “acid is corrosive” to solve a series of problems in various scenarios that rely on this fact.

---

> > ### Author Response · Authors · 2024-08-12
> > **Follow-up Rebuttal**
> >
> > ### Dear Reviewer Fwhs:
> >
> > We appreciate your thoughtful review. As the rebuttal deadline approaches, we kindly ask if our responses have sufficiently addressed your concerns. If you require further clarification, we are prepared to provide additional information.
> >
> > Sincerely,
> >
> > Authors

---

> > ### Comment · Reviewer_Fwhs · 2024-08-12
> >
> > Thanks for the clarification. I have looked through all the reviews and rebuttals and I will keep my rating positive for this paper.

---

### Official Review · Reviewer_zthW · 2024-07-12

**Soundness:** 2
**Presentation:** 1
**Contribution:** 2
**Rating:** 3
**Confidence:** 5

**Summary:**

This paper explores the abstract reasoning abilities of LLMs by creating a specific evaluation metric and a dataset called AbsR, developed using GPT-4. It presents a method, Meaningful Learning (MeanLearn), to improves both general and abstract reasoning accuracies of LLMs by teaching them to apply generic facts in varied scenarios.

**Strengths:**

- Proposing a dataset to study the potential of LLMs in abstract reasoning.
- Proposing MeanLearn to improve the abstract reasoning capabilities of LLMs by teaching them to use generic facts for reasoning.

**Weaknesses:**

- The paper is rushed, and not well organized. For instance, Some experimental results are presented in section 2, and others appear later in section 5 and after.
- The methodology section is half a page. More details on the model is needed. For instance, in Eq. 3, how the first and second equations will be utilized in the model.
- Incomplete related work section, specifically, insufficient related work on Abstract Reasoning.
- Some experimental results are in section 2: Abstract reasoning study, some later in section 5: Experiments. In section 2, the paper jumps into some results and experiments too early and without proper background.
- Experiments are limited. For example, it is unclear how the methods compare to [1]

- [1] Xu, Yudong, et al. "LLMs and the Abstraction and Reasoning Corpus: Successes, Failures, and the Importance of Object-based Representations." Transactions on Machine Learning Research.

**Questions:**

- How the methods introduced in the paper can be compare to [1]?
- The paper states that "Our findings reveal a substantial discrepancy between their general reasoning and abstract reasoning performances". However, based on [2], LLMs "know" more than they can "say". Please justify how your statement remains valid.
- Considering that MeanLearn does not have many advantages over LLaMA-3, how do you justified the use of MeanLearn?

- [1] Xu, Yudong, et al. "LLMs and the Abstraction and Reasoning Corpus: Successes, Failures, and the Importance of Object-based Representations." Transactions on Machine Learning Research.

- [2] Li, Kenneth, et al. "Inference-time intervention: Eliciting truthful answers from a language model." Advances in Neural Information Processing Systems 36 (2024).

**Limitations:**

- Lack of sufficient experiments.
- Limited related work, and background sections.
- A thorough writing revision would be beneficial.

---

> ### Author Rebuttal · Authors · 2024-08-02
>
> We would like to clarify some critical misunderstandings in the preliminary review, which may have negative impacts on your assessment of our contributions.
> # 1. The paper is rushed (Weakness 1)
> Both sections 2 and 5 contain results, and this arrangement is intentional and not rushed. As in many prior works [1][2], the results in section 2 are preliminary and are used to demonstrate our motivations and introduce our following method. Meanwhile, the results in section 5 are related to our proposed methods and baselines. Thus, the results in sections 2 and 5 are independent and serve different purposes within our paper.
>
> [1] Zhang et al., 2023: Automatic Chain of Thought Prompting in Large Language Models
>
> [2] Jin et al., 2024: The Impact of Reasoning Step Length on Large Language Models
> # 2. The methodology section is half a page (Weakness 2)
> Due to the page limit, we are trying to put more important content in the main paper. To accommodate as much as experimental results, we describe our approach as concisely as possible (More details can refer to Appendix C and D). It is worth noting that our method has two sections, the construction method of AbsR dataset (section 3) and MeanLearn pipeline (section 4), a totally of 1.5 pages.
>
> In Eq. (3), the first equation describes using input (X) to autoregressively model output (Y), the second is using X and generic fact r to autoregressively model Y, which is usually adopted by current LLMs posting-training works [3][4].
>
> [3] Orca 2: Teaching Small Language Models How to Reason
>
> [4] WizardLM: Empowering Large Language Models to Follow Complex Instructions
>
> # 3. Insufficient related work on Abstract Reasoning (Weakness 3)
> To the best of our knowledge, this is the first work to highlight the importance of abstract reasoning in the field of natural language processing. Thus, we include the most similar research directions: “Reasoning with LLMs” and “LLMs-as-annotators” in the related work section. Due to page limitations, we have condensed the related work. These sections will be expanded in the revised version.
>
> # 4. Comparison to paper [5] (Weakness 4 and Question 1)
> We believe there may be some critical misunderstandings concerning our paper and your referred paper. A comparison with this reference is unnecessary because: (1) it focuses on the reasoning on images or symbols, whereas our work is centered on abstract reasoning in natural language; (2) the paper does not propose a new method but merely presents an empirical study using ChatGPT and LLaMA on image tasks.
>
> [5] LLMs and the Abstraction and Reasoning Corpus: Successes, Failures, and the Importance of Object-based Representations
>
> # 5. The statement in our paper "Our findings reveal a substantial discrepancy between their general reasoning and abstract reasoning performances" and "LLMs know more than they can say" in [6] (Question 2)
> These two statements indeed do not conflict. Most researches follow the conventional evaluation process to directly evaluate LLMs based on their outputs, quantifying their reasoning abilities. Our proposed abstract reasoning metric AbsAcc based on cognitive theory, and our results show a substantial disparity between general and abstract reasoning. On the other hand, the statement “LLMs know more than they can say” may be derived from the domain of interpretability. It concerns the LLMs’ inference processes on tasks, which may not be in a form understandable to humans, leading to performance loss when translating these processes into natural language. The focal points of our work and the work you referred to are distinct, each addressing different aspects of LLM capabilities and behaviors.
>
> [6] Inference-time intervention: Eliciting truthful answers from a language model.
> # 6. the effectiveness of MeanLearn (Question 3)
> The benefits of our method MeanLearn do not diminish, but it is LLaMA-3 may need more data compared with other base models. The reason is because LLaMA-3 breaks conventional data scaling laws [7] by achieving a token-to-parameter (T/P) ratio of 1875:1, far surpassing the 286:1 ratio of LLaMA-2. This results in dense knowledge in the parametric memory of LLaMA-3. However, to ensure fair comparisons across different LLMs, we train MeanLearn based on LLaMA-3 with the same dataset size as the other LLMs. Furthermore, MeanLearn achieved satisfied improvements on LLaMA-2 and Orca-2.
>
> [7] Pearce and Song, 2024. Reconciling Kaplan and Chinchilla Scaling Laws

---

> > ### Author Response · Authors · 2024-08-12
> > **Follow-up Rebuttal**
> >
> > ### Dear Reviewer zthW:
> >
> > We appreciate your efforts in the review process. As the rebuttal deadline approaches, we kindly ask if our responses have sufficiently addressed your concerns. If you require further clarification, we are prepared to provide additional information.
> >
> > Sincerely,
> >
> > Authors

---

> > > ### Comment · Reviewer_zthW · 2024-08-13
> > >
> > > Q6. The authors have not responded to my concern clearly. The effectiveness of MeanLearn compared to the Llama-3 8B model is unclear. While the authors mention the computational cost of Llama compared to other LLMs, they have not justified how MeanLearn's performance compares to Llama-3. Additionally, the lack of evaluation against Llama-3 13B raises concerns about how MeanLearn is effective on larger Llama-3 13B model (While some other 13B models have been tested).
> > >
> > > Q5. The authors argument about not including other baseline is not convincing (as other reviewers have noted that). Thus, the contribution and the benefits of the work is not clear.
> > >
> > > Thanks to the authors for their responses, I decided to keep my score.

---

> ### Author Response · Authors · 2024-08-13
> **Replying to Official Comment by Reviewer zthW**
>
> Thanks for your reply. We want to clarify the critical misunderstandings in your response:
>
> ## 1. Q6 (the author should conduct experiments on LLaMA-3-13B) in the response
>  **We do not incorporate LLaMA-3-13B because there is NO 13B version of LLaMA-3.** (refer to https://huggingface.co/collections/meta-llama/meta-llama-3-66214712577ca38149ebb2b6) The second smallest LLaMA-3 is 70B. We indeed do not have the resources for training and inference. By the way, our MeanLearn method is effective on LLaMA-2-7B, LLaMA-2-13B, LLaMA-3-8B, Orca-2-7B, Orca-2-13B, and Mistral-7B (as mentioned in response to Reviewer XVL7, and you can refer to the table below).
>
> | Method                 | AbsR  | Comm. | MMLU  | RACE  |     *&nbsp;&nbsp;&nbsp;&nbsp;&nbsp;Average*     |
> | ---------------------- | :---: | :---: | :---: | :---: | :---------------: |
> | ***Vanilla Accuracy*** |       |       |       |       |                   |
> | Mistral-7B-Instruct    | 83.00 | 66.53 | 57.02 | 75.33 |       70.47       |
> | MeanLearn (Ours)       | 84.50 | 74.56 | 58.28 | 76.65 | **73.50 (+3.03)** |
> | ***AbsAcc***           |       |       |       |       |                   |
> | Mistral-7B-Instruct    | 72.92 | 58.87 | 40.19 | 52.69 |       56.17       |
> | MeanLearn (Ours)       | 73.96 | 68.59 | 41.14 | 55.77 | **59.87 (+3.60)** |
>
> About the performance on LLaMA-3-8B, we have offered some clarification in the rebuttal period, which is summarized as follows:
> * We have improvements on LLAMA-3-8B;
> * LLaMA-3 may need more data compared to other base models. The reason is that LLaMA-3 breaks conventional data scaling laws [1] by achieving a token-to-parameter (T/P) ratio of 1875:1, far surpassing the 286:1 ratio of LLaMA-2. This results in dense knowledge in the parametric memory of LLaMA-3. However, to ensure fair comparisons across different LLMs, we trained MeanLearn using LLaMA-3 with the same dataset size as the other LLMs.
> * Although incorporating more data to train LLaMA-3 is unfair for the comparisons with other LLMs, we are interested in adopting more data to train LLaMA-3 in the future. Due to time limits and computational cost, we do not provide it in the rebuttal.
>
>
>
>
>
> ## 2. Q5 in the response
>
> * For your referred paper[2], we do not incorporate it for comparison because: (1) it is a paper of abstraction AND reasoning on **Visual Inputs** and (2) it is an **empirical study** without proposing new methods or datasets. While we focus on abstracting reasoning on **Natural Language**.
>
> * For the baseline proposed by Reviewer vYe8, we do not incorporate tree-of-thought (tot) for comparison because: tot focuses on the **decoding stage**, while we focus on the **post-training stage** to enhance the fundamental capabilities, they are **complementary**. We adhere to conventional standards by selecting baselines of similar types and scales to ensure a fair evaluation.
>
> * Both Reviewers XVL7 and Fwhs think the soundness of our work is good.
>
> [1] Pearce and Song, 2024. Reconciling Kaplan and Chinchilla Scaling Laws
>
> [2] Xu, Yudong, et al. "LLMs and the Abstraction and Reasoning Corpus: Successes, Failures, and the Importance of Object-based Representations." Transactions on Machine Learning Research.

---

### Official Review · Reviewer_vYe8 · 2024-07-13

**Soundness:** 2
**Presentation:** 3
**Contribution:** 2
**Rating:** 5
**Confidence:** 3

**Summary:**

This paper introduces a novel framework aimed at enhancing the abstract reasoning capabilities of large language models (LLMs) through a method called "Meaningful Learning." It specifically targets the challenge LLMs face in abstract reasoning despite their robust general reasoning abilities. The authors identify a notable gap in performance between general and abstract reasoning tasks and propose a structured approach to narrow this gap by using AbsR, a tailored dataset that includes generic facts coupled with guided explanations to foster deeper learning and understanding.
Key contributions include:
1. Introduction of the Meaningful Learning framework for improving abstract reasoning in LLMs.
2. Development of the AbsR dataset for training.
3. New metrics and Empirical evaluation across different settings

**Strengths:**

1. Novel approach to improving abstract reasoning in LLMs through generic fact guidance.
2. Comprehensive evaluation across multiple settings demonstrating reasonable performance improvements.

**Weaknesses:**

1. Limited scale: The experiments are conducted on relatively small models (7B-13B parameters) compared to state-of-the-art LLMs. Without such it's hard to judge such method's applicability in real world and more complex tasks as mentioned in 5.1.
2. Human evaluation scale: The human evaluation of the AbsR dataset is conducted on a relatively small sample (200 instances) (Appendix E).
3. Lack of comparison to more recent reasoning techniques: The paper doesn't compare MeanLearn to recent advances in LLM reasoning capabilities.
4. Weaknesses in Evaluation Metrics: It's not convincing that Perplexity-based evaluation for classification tasks is good and the use of AbsAcc needs more clarification.

**Questions:**

1. How did you determine that perplexity-based evaluation was the best approach for classification tasks, given its limitations mentioned in Appendix A? Were other methods considered?
2. Have you considered comparing MeanLearn to more recent reasoning techniques, such as tree-of-thoughts or other advanced prompting methods
3. The paper mentions that "MeanLearn can make LLMs implicitly learn generic facts and solve problems under the guidance of generic facts" (Section 4). Can you provide more concrete examples or analysis of how this implicit learning occurs?
4. How do you envision MeanLearn scaling to larger models (100B+ parameters), and what challenges or benefits do you anticipate in applying this method to more advanced LLMs?
5. how do you ensure that the AbsAcc metric is truly capturing abstract reasoning abilities rather than other factors like memorization or pattern matching?

**Limitations:**

Yes

---

> ### Author Rebuttal · Authors · 2024-08-02
>
> Thanks for your insightful comments, the following is the detailed response:
> # 1. Limited scale (Weakness 1 and Question 4)
> ## 1.1 Limited scale
> Due to limited computational resources, we conduct our experiments on small LLMs (7B-13B). It is worth noting that our main focus is also to improve small LLMs, which is also crucial, as evidenced by considerable research focus in this area [1][2]. Small LLMs have a broader range of applications and are preferred for deployment on devices like cell phones due to their lower resource demands. Despite their wider applicability, small LLMs are significantly weaker than their larger counterparts, underscoring the importance of prioritizing their improvement.
>
> [1] Orca 2: Teaching Small Language Models How to Reason
>
> [2] WizardLM: Empowering Large Language Models to Follow Complex Instructions
> ## 1.2 Applying MeanLearn to 100B+ LLMs
> We are attempting to discuss to apply our method to 100B+ LLMs, but it is out of the scope of this paper. Applying MeanLearn to 100B+ LLMs would presents both challenges and benefits:
> Challenges: (1) Resource Requirements: the post-training of such large LLMs demands additional and more stable computational resources to manage the high costs associated with their scale; (2) Data Demands: according to the data scaling law [3], there is a significant need for extensive training data to achieve adequate coverage and ensure the generalization capabilities of these large models. Acquiring such large datasets is often costly.
> Benefits: (1) Enhanced Learning Abilities: larger LLMs possess stronger learning capacities, enabling them to more effectively comprehend and assimilate the knowledge presented to them; (2) Superior Performance: these models typically outperform smaller LLMs, offering better overall performance in tasks due to their advanced capabilities.
>
> [3] Pearce and Song, 2024. Reconciling Kaplan and Chinchilla Scaling Laws
> # 2. Human evaluation scale (Weakness 2)
> We mainly follow previous works [4] [5], which conduct human evaluation on 100-200 instances. We choose 200 instances for evaluation, which is a balance between maintaining evaluation quality and managing labor costs.
>
> [4] Du et al., e-CARE: a New Dataset for Exploring Explainable Causal Reasoning
>
> [5] Ying et al., Intuitive or Dependent? Investigating LLMs' Robustness to Conflicting Prompts
> # 3. Lack of comparison to more recent reasoning techniques (Weakness 3 and Question 2)
> We adhere to conventional standards by selecting baselines of similar types and scales to ensure a fair evaluation. Our focus is on enhancing the fundamental capabilities of LLMs through post-training techniques. In contrast, methods like Tree-of-Thought (ToT) primarily target improvements during the decoding stage, which complements our proposed method.
> # 4. Perplexity-based evaluation (Weakness 4 and Question1)
> We align with the evaluation criteria established by prior research [6][7] and leaderboards [8][9], adopting perplexity as our evaluation metric. The advantages of employing perplexity are twofold:
> * Clarity in Evaluation: evaluation with perplexity can extract an answer for each example, while generation-based evaluation is hard to do this, since LLMs might either refuse to provide an answer or treat any option as valid;
> * Efficiency: Calculating perplexity requires only a single forward pass through the LLMs, making it more cost-efficient compared to generation-based methods, which require text to be generated in an autoregressive manner. This streamlined process enhances both the speed and the resource efficiency of the evaluation.
>
> [6] Sun et al., 2024: A Simple and Effective Pruning Approach for Large Language Models
>
> [7] Zhao et al., 2024: Deciphering the lmpact of Pretraining Data on Large Language Models through Machine Unlearning
>
> [8] OpenCompass, 2023: OpenCompass: A Universal Evaluation Platform for Foundation Models
>
> [9] Huggingface 2024: Open LLM Leaderboard
> # 5. About AbsAcc (Weakness 4 and Question 5)
> Theoretically, AbsAcc offers a more reliable measurement of abstract reasoning. For each generic fact, we have multiple examples to test LLMs, effectively reducing the influences of memorization and pattern matching. Ideally, increasing the number of examples for each generic fact would enhance the reliability of AbsAcc. We do not incorporate more examples per generic fact for the balance between cost and effectiveness. Please note that, our method is scalable with respect to the number of examples for each generic fact.
> # 6. Examples or analysis of of the implicit learning process (Question 3)
> Humans can apply the grasped generic fact to solve problems in different scenarios. For example, in Figure 1 of our paper, if humans grasp “acid is corrosive”, they can deduce “rock dissolved” and “the skin suffers pain” when respectively given “adding rock into hydrochloric acid” and “acid touches human skin”. MeanLearn is designed to imitate the above meaningful learning process of humans.
>
> As for analysis, this question is indeed interesting and merits exploration. However, it falls outside the scope of the main contributions of our work. Our focus is not on demonstrating the implicit learning processes within a black-box LLM, as that pertains more to the field of interpretability. Addressing this would require a significantly different research approach and substantial additional effort.

---

> ### Author Response · Authors · 2024-08-12
> **Follow-up Rebuttal**
>
> ### Dear Reviewer vYe8:
>
> We appreciate your thoughtful review. As the rebuttal deadline approaches, we kindly ask if our responses have sufficiently addressed your concerns. If you require further clarification, we are prepared to provide additional information.
>
> Sincerely,
>
> Authors

---

> > ### Comment · Reviewer_vYe8 · 2024-08-13
> >
> > Thanks for the clarification. I have looked through all the reviews and am less concerned about the evaluation; Yet I think there is still a lack of comparison with existing post-training methods which should have been detailed in the related work section, I will keep my rating neutrally positve for this paper.

---

> ### Author Response · Authors · 2024-08-13
>
> Thanks for your response, we are glad to have addressed your concerns in your preliminary review, and we want to kindly remind that:
>
> Actually, **in Table 5 of our paper, we have provided the post-training baselines** (Vicuna [1], WizardLM [2], Orca-2 [3]) , and you can refer to the table below (the average performance on 7B models), **we have varying advantages over these methods on both vanilla accuracy and AbsAcc**.
>
> | Method           | *Average* |
> |------------------|:-------:|
> | ***Vanilla Accuracy*** |         |
> | Vicuna           |  56.48  |
> | WizardLM         |  31.28  |
> | Orca-2           |  61.84  |
> | MeanLearn (Ours) |  **65.28**  |
> | ***AbsAcc***           |         |
> | Vicuna           |  39.84  |
> | WizardLM         |  20.42  |
> | Orca-2           |  48.86  |
> | MeanLearn (Ours) |  **53.39**  |
>
> If you have any further concerns, feel free to leave comments.
>
> [1] https://lmsys.org/blog/2023-03-30-vicuna/
>
> [2] WizardLM: Empowering Large Language Models to Follow Complex Instructions
>
> [3] Orca 2: Teaching Small Language Models How to Reason

---

### Official Review · Reviewer_XVL7 · 2024-07-14

**Soundness:** 3
**Presentation:** 2
**Contribution:** 2
**Rating:** 6
**Confidence:** 3

**Summary:**

This paper addresses the challenge that LLMs face in abstract reasoning, where they often struggle to apply general facts to new situations despite their impressive performance in other reasoning tasks. To tackle this issue, the authors introduce an abstract reasoning dataset called AbsR, which incorporates generic facts and guided explanations to teach LLMs how to leverage such facts for reasoning. They also propose a learning paradigm named Meaningful Learning (MeanLearn) that simulates the human process of implicit knowledge acquisition, enabling LLMs to implicitly learn and apply generic facts without explicit input. Through experiments on various out-of-distribution reasoning and language understanding benchmarks, the paper demonstrates that MeanLearn improves the general and abstract reasoning capabilities of LLMs, moving beyond simple memorization towards a more nuanced understanding and application of knowledge.

**Strengths:**

1. Useful Resource: This paper introduces an abstract reasoning dataset (AbsR) that provides generic facts and guided explanations for reasoning tasks. The dataset and code will be publicly available, facilitating reproducibility and further community research.
2. Empirical Evidence: Comprehensive experimental results and ablation studies that validate the effectiveness of the proposed method.  Improvements are observed in both general and abstract reasoning performance of LLMs across various benchmarks.
3. Broad Applicability: The approach's effectiveness is shown on multiple LLMs of varying sizes, indicating broad applicability.

**Weaknesses:**

1. Evaluation on Advanced LLMs: The observation that the benefits of MeanLearn seem to diminish with better-trained LLMs, like LLaMA3, raises a crucial question about its broader applicability. While the method shows promise with smaller models, it's essential to assess its performance on more powerful LLMs like Mistral, Phi3, and Qwen 2. This would provide a clearer picture of its potential contribution in the context of rapidly advancing language models.

2. Data Augmentation and Training Dynamics: The use of GPT4 for constructing the AbsR dataset, with annotation quality comparable to human annotators, opens up interesting possibilities for data augmentation. Exploring the impact of increasing the training data size, potentially using more cost-effective models like GPT4o or open-source LLMs, could reveal the potential for further performance gains. Additionally, analyzing the training dynamics of MeanLearn by varying the proportion of training data (e.g., 20%, 50%, 80%) would provide valuable insights into its saturation point and the diminishing returns of additional data.

3. Expanding Reasoning Benchmarks: The paper's focus on commonsense reasoning benchmarks (Com. and ARC) should be complemented by evaluation on other reasoning domains, such as arithmetic reasoning. Including benchmarks like GSM8K or MATH would provide a more comprehensive understanding of MeanLearn's capabilities across different reasoning tasks.

4. Presentation Clarity and Improvements: The paper's presentation could benefit from several improvements. The introduction should explicitly connect abstract reasoning with other types of reasoning, providing a broader context for the research. The pilot investigation in the introduction is quite confusing as it does not introduce the datasets, metrics, and setup used. Confusing figures, like Figure 2, should be explained in detail to ensure clarity and understanding.

5. Updated Related Work: The related work section should be updated to include recent efforts in tuning-based methods, ensuring a comprehensive overview of the current landscape in abstract reasoning research.

**Questions:**

See above,

**Limitations:**

The authors adequately addressed the limitations.

---

> ### Author Rebuttal · Authors · 2024-08-02
>
> Thanks for your insightful comments, the following is the detailed response:
> # 1. Evaluation on Advanced LLMs (Weakness 1)
> The benefits of our method MeanLearn do not diminish, but it is LLaMA-3 may need more data compared with other base models. The reason is because LLaMA-3 breaks conventional data scaling laws [1] by achieving a token-to-parameter (T/P) ratio of 1875:1, far surpassing the 286:1 ratio of LLaMA-2. This results in dense knowledge in the parametric memory of LLaMA-3. However, to ensure fair comparisons across different LLMs, we trained MeanLearn using LLaMA-3 with the same dataset size as the other LLMs.
>
> Due to limited time, we train MeanLearn based on Mistral-7B-Instruct-v0.2, and evaluate them on Com., MMLU, RACE and AbsR, the results are shown in the following table:
>
> |Method|AbsR|Com.|MMLU|RACE|*&nbsp;&nbsp;&nbsp;&nbsp;&nbsp;Average*|
> |---------------------|:-----:|:-----:|:-----:|:-----:|:---------:|
> |***Vanilla Accuracy***||||||
> |Mistral-7B-Instruct|83.00|66.53|57.02|75.33|70.47|
> |MeanLearn(Ours)|84.50|74.56|58.28|76.65|**73.50 (+3.03)**|
> |***AbsAcc***||||||
> |Mistral-7B-Instruct|72.92|58.87|40.19|52.69|56.17|
> |MeanLearn(Ours)|73.96|68.59|41.14|55.77|**59.87 (+3.60)**|
>
> On average, our proposed MeanLearn can outperform Mistral by 3.03% in vanilla accuracy and 3.70% in AbsAcc, which demonstrates the superiority of MeanLearn. The main reason is that Mistral does not have a dense knowledge in parameters as LLaMA-3.
>
> [1] Pearce and Song, 2024. Reconciling Kaplan and Chinchilla Scaling Laws
> # 2. Data Augmentation and Training Dynamics (Weakness 2)
> GPT-4o was released about a week before the NeurIPS 2024 deadline, making it too late to use for synthesizing AbsR. Indeed, leveraging cost-effective models like GPT-4o or open-source LLMs for data generation is crucial for future performance improvements. We are enthusiastic about exploring this and the training dynamics in the future to achieve continual performance gains.
> # 3. Expanding Reasoning Benchmarks (Weakness 3)
> There are mathematical tasks in MMLU, following [2], we select five mathematical reasoning datasets (abstract algebra, college mathematics, elementary mathematics, high school statistics, and high school mathematics) from MMLU to demonstrate the superiority of MeanLearn. The results are shown in the following table:
> |Size|Method|Abstract Algebra|College Mathematics|Elementary Mathematics|High School Statistics|High School Mathematics|*Average*|
> |------------------|-----------|:----------------:|:-------------------:|:----------------------:|:----------------------:|:-----------------------:|:-------------:|
> |***Vanilla Accuracy***||||||||
> |7B|Orca-2|23.00|32.00|33.07|37.95|28.15|30.83|
> ||MeanLearn (Ours)|36.00|35.00|33.07|40.28|28.89|**34.65 (+3.82)**|
> ||Mistral|25.00|30.00|38.89|46.76|32.96|34.72|
> ||MeanLearn (Ours)|31.00|34.00|37.83|47.69|34.07|**36.92 (+2.20)**|
> |8B|LLaMA-3|30.00|36.00|41.27|50.00|40.74|39.60|
> ||MeanLearn (Ours)|31.00|33.00|44.44|49.54|40.74|**39.74 (+0.14)**|
> |13B|Orca-2|27.00|35.00|35.45|34.72|28.15|32.06|
> ||MeanLearn (Ours)|27.00|37.00|37.30|43.06|27.78|**34.43 (+2.37)**|
> |***AbsAcc***||||||||
> |7B|Orca-2|0.00|4.88|25.00|25.17|14.63|13.94|
> ||MeanLearn (Ours)|2.86|7.32|24.02|27.97|16.26|**15.69 (+1.75)**|
> ||Mistral|2.86|7.32|25.49|36.36|14.63|17.33|
> ||MeanLearn (Ours)|5.71|9.76|24.55|37.76|16.26|**18.81 (+1.48)**|
> |8B|LLaMA-3|2.86|9.76|28.43|38.46|18.70|19.64|
> ||MeanLearn (Ours)|5.71|12.20|30.88|34.97|19.51|**20.65 (+1.01)**|
> |13B|Orca-2|2.86|4.88|24.02|24.48|13.82|14.01|
> ||MeanLearn (Ours)|2.86|17.07|22.55|29.37|10.57|**16.48 (+2.47)**|
>
> On average, MeanLearn outperforms baselines on both vanilla accuracy and AbsAcc. It is interesting to note that we do not synthesize math questions to train MeanLearn, improvements on math reasoning are largely due to the enhancement of abstract reasoning.
>
> Meaningful Learning is one of the main contributions of our work, and we are excited about the improvements brought by MeanLearn. Building on this foundation, we are now conducting a new work to expand our investigations with additional tasks such as mathematical and logical reasoning.
>
> [2] MMLU-Pro: A More Robust and Challenging Multi-Task Language Understanding Benchmark
>
> # 4. Presentation Clarity and Improvements (Weakness 4)
> Abstract reasoning is not a specific type of reasoning; rather, it measures the ability of employing generic facts to solve problems. For each reasoning type, an abstract reasoning performance metric can be derived. For example, in math problems, abstract reasoning can help estimate the ability of LLMs to use the generic fact “x + y = y + x” to solve a series of problems in various scenarios that rely on this fact. While in commonsense problems, it can help estimate the ability of LLMs to use the generic fact “acid is corrosive” to solve a series of problems in various scenarios that rely on this fact.
>
> The pilot investigation results presented in introduction serve to intuitively highlight the disparity (Table 1) between general and abstract reasoning, which is used to demonstrate our motivation that LLMs have a substantial discrepancy between their general reasoning and abstract reasoning performances. Detailed discussions of this investigation are available in sections 2.1 and 2.3.
>
> Furthermore, Figure 2 illustrates the calculations for vanilla accuracy and AbsAcc, employing the same symbols and formulas outlined in section 2.1 and equation (1).
>
> # 5. Updated Related Work (Weakness 5)
> Due to page limitations and to place more experimental results, we have condensed some sections of the related work. We will expand the related work in the revised version.

---

> > ### Comment · Reviewer_XVL7 · 2024-08-09
> >
> > Thanks for your response. The experimental results addressed some of my concerns. I am willing to increase my rating for this paper, but I would like to ask one more question. Why do you use MMLU to test the mathematical reasoning abilities instead of GSM8K and MATH?

---

> > > ### Author Response · Authors · 2024-08-09
> > > **Replying to Comment by Reviewer**
> > >
> > > Thanks for your comment. We are glad to have addressed some of your concerns. We choose the mathematical reasoning tasks in MMLU for experiments mainly due to:
> > > * MMLU has a more fine-grained classification of the tasks, such as algebra and statistics. By using this, we can obtain a more comprehensive evaluation of the methods;
> > > * Previous researches [1][2] utilize this subset to evaluate the mathematical capabilities of LLMs;
> > > * We have already conducted experiments on MMLU in our submission, due to time limitations, we can quickly obtain the mathematical reasoning results from previous experiments.
> > >
> > > We hope this answers your question. Moreover, we are interested in incorporating more datasets like MATH and GSM8K for further investigation, and this is what we are doing in our new work.
> > >
> > > If you have any further questions or concerns, feel free to leave comments.
> > >
> > > [1] Want et al., 2024. Mmlu-pro: A more robust and challenging multi-task language understanding benchmark
> > >
> > > [2] Luo et al., 2023. Wizardmath: Empowering mathematical reasoning for large language models via reinforced evol-instruct

---

> ### Comment · Reviewer_XVL7 · 2024-08-09
>
> Thanks for the clarification. I would like to mention that MATH also has a fine-grained classification for analysis. I will keep my rating positive for this paper.

---

### Decision · Program_Chairs · 2024-09-25

**Decision:**

Accept (poster)

**Comment:**

This paper represents a relatively substantial amount of work about reasoning in LLMs. While one reviewer believes there are issues with the paper to be resolved, it seems like the overall consensus is that the paper represents sufficient amount of work. I ask that the authors work specifically on the representational issues that are mentioned by the reviewers.